# Advances in Computational Techniques for Bio-Inspired Cellular Materials in the Field of Biomechanics: Current Trends and Prospects

**DOI:** 10.3390/ma16113946

**Published:** 2023-05-25

**Authors:** A. I. Pais, J. Belinha, J. L. Alves

**Affiliations:** 1Institute of Science and Innovation in Mechanical and Industrial Engineering (INEGI), 4200-465 Porto, Portugal; ana.i.pais@gmail.com (A.I.P.); falves@fe.up.pt (J.L.A.); 2Department of Mechanical Engineering, ISEP, Polytechnic University of Porto, 4200-465 Porto, Portugal; 3Department of Mechanical Engineering, FEUP, University of Porto, 4200-465 Porto, Portugal

**Keywords:** artificial intelligence, machine learning, deep learning, computational methods, scaffold, cellular materials

## Abstract

Cellular materials have a wide range of applications, including structural optimization and biomedical applications. Due to their porous topology, which promotes cell adhesion and proliferation, cellular materials are particularly suited for tissue engineering and the development of new structural solutions for biomechanical applications. Furthermore, cellular materials can be effective in adjusting mechanical properties, which is especially important in the design of implants where low stiffness and high strength are required to avoid stress shielding and promote bone growth. The mechanical response of such scaffolds can be improved further by employing functional gradients of the scaffold’s porosity and other approaches, including traditional structural optimization frameworks; modified algorithms; bio-inspired phenomena; and artificial intelligence via machine learning (or deep learning). Multiscale tools are also useful in the topological design of said materials. This paper provides a state-of-the-art review of the aforementioned techniques, aiming to identify current and future trends in orthopedic biomechanics research, specifically implant and scaffold design.

## 1. Introduction

The use of scaffold-based bone tissue engineering in the future has considerable potential for treating large bone defects. Some current examples in orthopedic biomechanics can be seen for example in the work of Blázquez-Carmona et al. [1] who, to ensure focused adhesion between the cell and the biomaterial, as well as cell growth and differentiation, employed a cost function in optimization based on biological factors to develop patient-specific ceramic implants with optimal geometric parameters. Jardini et al. [2] presents a case-study of a personalized titanium (Ti-6Al-4V) cranial plate manufactured through direct metal laser sintering (DMLS). Zhang et al. [3] developed patient-specific implants and in this study, the β-TCP bone scaffold provides proof that critical-sized craniomaxillofacial bone defects can be repaired and replaced using digital light processing (DLP) technology.

Although revision procedures are frequently required and performed, bone implants remain as one of the riskiest medical devices. The most frequent reason for revision surgery is implant loosening brought on by bone resorption, resulting from the stress shielding effect [4].

The 3D matrix known as a bone scaffold is what enables and promotes osteoinducible cells to connect to and proliferate on its surfaces [5]. The main requirements in scaffold design can be summarized as the following [5]:biocompatibility;biodegradability;mechanical properties to bear weight during the amelioration period;proper architecture in terms of porosity and pore sizes;sterilibility without loss of bioactivity;controlled deliverability of bioactive molecules or drugs.

When it comes to mechanical properties, the design of the bone scaffold can be broken down into two main categories: strength to resist the loads that naturally occur in the human body and the scaffold’s stiffness, which must be kept at a minimum to allow for sufficient load transfer to the bone. Bone does exhibit anisotropy, but there are no extremely weak directions, so it is adequate if its mechanical properties are similar to those of the nearby bones on average [6,7,8,9].

Porosity management is a crucial factor that must be taken into consideration when designing implants. For the device to be successful, it is essential to control the porosity, the pore size, and the pore interconnectivity. Higher porosity promotes vascularization and the recruitment of cells from the nearby tissue, which helps bone ingrowth [6,10]. Because various references identify different pore sizes as being ideal for bone ingrowth, the influence of pore size is not as widely agreed upon among researchers [6,11,12,13]. The specific surface area of the scaffold is also very important because bigger areas for bone tissue ingrowth result from higher surface areas, which are more likely to occur in scaffolds with smaller pores [6,14]. Nevertheless, a greater surface area entails greater frictional forces, which hinder permeability [7]. Permeability is important in the construction of the scaffold because the transportation of cells, nutrients, and growth factors requires a blood flood in the scaffold [6,7].

### 1.1. Meta-Materials

Meta-materials can optimize the specific material properties, which has led to their development and research. It is possible to develop materials with high stiffness-to-weight and strength-to-weight ratios [15]. By altering the unit cell geometry, the selected function can be improved. Additionally, through gradient structures [16,17,18] or hierarchical structures [19,20], the macroscale properties can be tuned to fit some purpose. Some examples of the efficiency of such materials are found in nature; for example, bone, which is a hierarchical structure with naturally occurring density gradients for the optimization of stiffness. Cellular materials, used in tissue engineering, can be considered to be meta-materials in the sense that the properties are customized in order to achieve a specific function, by altering the unit cell geometry for example. It is possible to accurately tune the properties of the material at the macroscale by designing the material at the lower length scale. Therefore, these materials tend to be analyzed at two length scales, the macro scale and the micro-scale, via homogenization techniques.

The literature presents several examples of cellular materials used in tissue engineering. These can be divided into three categories, as shown in Figure 1, truss-like (or CAD-based), TPMS, or stochastic. TPMS stands for triply periodic minimal surfaces and, thus, the material structure is defined by the parameters in the function equation. Further, an additional category can be included, which is topology optimization-based, i.e., the unit cell geometry is obtained from topology optimization analysis [21]. However, the obtained geometries will often appear TPMS-like or truss-like. Stochastic scaffolds can be obtained from image segmentation or through random design, for example with the Voronoï tesselation.

Within the TPMS classification, the structures can also be divided between sheet-like or strut-like, even when the base surface is the same for both cases. This distinction depends on how the surface is turned into a solid structure. One known TPMS is the schoen gyroid, as shown in Equation (Equation 1).
(1)$$F:\;sin\left(\frac{2\pi x}{L}\right)cos\left(\frac{2\pi y}{L}\right)+sin\left(\frac{2\pi y}{L}\right)cos\left(\frac{2\pi z}{L}\right)+sin\left(\frac{2\pi z}{L}\right)cos\left(\frac{2\pi x}{L}\right)-t=0$$
where *L* is the unit cell size and parameter *t* translates the equation so that if *t* is equal to zero, the surface divides a cubic space into two equal volumes.

The creation of a strut-like solid or sheet-like solid can be characterized as follows:(2)$$\left\{\begin{array}{c} F<t_{1}\;\left(\mathrm{strut}-\mathrm{like}\right)\\ t_{1}<F<t_{2}\;\left(\mathrm{sheet}-\mathrm{like}\right) \end{array}\right.$$

TPMS structures, especially sheet-like ones, present one main advantage: their high surface area. As shown in (Equation 2), a sheet-like configuration consists of two gyroid surfaces and, thus, it presents double the surface area. For this reason, a gyroid sheet-like foam material is often called a double gyroid. Figure 2 shows the difference between strut-like and sheet-like TPMS.

### 1.2. Bone

Bone is a highly hierarchical structure (Figure 3) composed of both organic materials, namely collagen and fibrillin, and inorganic materials such as hydroxyapatite (HA). Each hierarchical level performs several functions of a mechanical, biological, and chemical nature [6].

The mechanical properties of bone depend on several factors, such as health, age, and location in the body. A summary of values found in the literature is shown in Table 1. Cortical bone is stronger and stiffer in the longitudinal direction than in the transverse direction, and stronger in compression than it is in tension. Trabecular bone is porous and anisotropic, being that its mechanical properties depend on the apparent density and trabecular orientation [6].

### 1.3. Additive Manufacturing

Although the complexity of shapes presented by scaffolds enables them to achieve optimal mechanical and biological properties, such complexity is also a limitation, since it hinders its use with conventional manufacturing approaches. Additive manufacturing (AM), by having as its basic idea the layer-by-layer building of the part, allows manufacturing the complex shape of bone scaffold. Table 2 presents a summary of the ASTM 52900 designation of AM technologies.

This section presents the technologies used in the manufacturing of bone scaffold. The most common materials used in porous bone scaffold implants are titanium/magnesium alloys, PEEK, and bio-ceramics [24].

#### 1.3.1. Metals

Metallic materials, for a long time, have been used in bone scaffolds and implants. Some examples include stainless steel, cobalt alloys, and titanium [6]. 316L stainless steel enjoys a wide use in the fabrication of scaffolds, as it presents superior corrosion resistance, bio-compatibility, and low cost [26]. Titanium alloys, such as Ti-6Al-4V, are also widely used due to their high strength-to-weight ratio and bio-compatibility [27]. Additionally, Zn-Mg alloys are also interesting because they exhibit favorable biodegradability properties in comparison to other materials such as iron (Fe), in which the degradation rate is slow, and magnesium (Mg), in which the degradation rate is too fast. Nevertheless, it is possible to tune the mechanical properties of Mg to match those of bone [28,29]. For instance, the literature shows that is possible to use an iron-manganese (Fe-Mn) alloy with silver (Ag) to accelerate degradation and increase anti-bacterial activity [30]. Some material properties, such as biocompatibilty, mechanical strength, and corrosion resistance, are superior in metals. However, the high stiffness those materials present are incompatible with bone, leading to stress shielding, which highlights the importance of topological design in order to adjust the stiffness of these bulk materials [6]. Some additive manufacturing (AM) processes for metals include selective laser melting (SLM), electron beam melting (EBM), direct metal laser sintering (DMLS), directed energy deposition (DED), and metal fused deposition modeling (FDM). The most common AM process to manufacture metallic implants is SLM. Table 3 shows a collection of works found in the literature using several materials, manufacturing processes, and topologies of scaffold implants.

#### 1.3.2. Polymers

The literature also shows some works considering the development of polymeric scaffolds. Here, extrusion processes such as extrusion litography/projection-based processes emerge, for example, FDM, SLA, and DLP. Some of the employed materials include PLA, which is biocompatible and biodegradable, as well as being easily processed through additive manufacturing. Some mechanical properties obtained from studies available in the literature are shown in Table 4. In general, polymeric scaffolds can exhibit mechanical properties matching trabecular bone, as well as being biodegradable and bio-compatible [23].

#### 1.3.3. Ceramics

Besides collagen, the other main material present in bones is hydroxiapatite, which is a ceramic material. Ceramic materials are interesting for bone scaffolding, as they are less stiff than metals, which can aid in the prevention of stress-shielding [4].

Some works using ceramic materials are the works of Vijayavenkataraman et al. [4] with alumina (Al_2_O_3_) TPMS scaffolds through litography-based ceramics manufacturing (LCM), where the obtained functionally graded scaffolds presented optimized properties capable of replacing trabecular bone and reducing stress shielding. In the work of Jiao et al. [24], Voronoi zirconia (ZrO_2_)/HA scaffolds manufactured using ceramic digital light processing (DLP) were developed and produced. Furthermore, in the work of Zhao et al. [42], HA reinforced with a small amount of graphene oxide (up to 0.4%) and ceramic DLP was developed and used.

The extrusion process of based HA+β-tricalcium phosphate (β-TCP) scaffolds was simulated by Bagwan et al. [43], and in the work of Elsayed et al. [44], borosilicate glass was used and scaffolds were printed using both DLP and direct ink writing (DIW). This last procedure allowed the gyroid scaffolds to achieve the highest compressive strength. Through DLP technology, in the research work of Liu et al. [45], HA scaffolds were printed, achieving biocompatibility and final mechanical properties highly compatible with cancellous bone.

Table 5 shows a summary of the achieved mechanical properties of the scaffolds presented in the mentioned works. Although some developed materials/scaffolds present adequate strength, they also show elevated stiffness. It is important that both properties match the properties of bone.

#### 1.3.4. Composites

Scaffolds may also be constituted of several other materials. Bone tissue is composed of collagen and HA, so the use of composite materials can increase aid in further mimicking the properties of bone with the scaffold. For example, the integration of soft tissue into PEEK implants can be enhanced through the addition of ceramic fillers, such as β-TCP or HA [51]. Metallic fillers can be added to polymer scaffolds in order to achieve a strengthening effect, as well as other biological characteristics such as cell viability and osteogenic, angiogenic, and anti-bacterial properties [52]. A summary of some approaches found in the literature is shown in Table 6.

## 2. Artificial Intelligence, Machine Learning, and Deep Learning

The terms machine learning and artificial intelligence are often used interchangeably. In fact, these are subsets within themselves (Figure 4). Table 7 shows the difference between the three terms.

Artificial intelligence (AI) refers to a broader “umbrella term”, which is the concept of creating smart intelligent machines, that is, to have the machine imitate human behavior. Hard coding a task could be considered artificial intelligence, since all possible cases are presented to the machine.

Therefore, machine learning (ML) surges as a subset of AI, allowing the system to automatically learn and improve from experience (Figure 5). ML is, thus, the discipline of computer science that uses computer algorithms and analytics to build predictive models. It uses large sets of data in order to learn patterns and rules from the data, using several techniques and algorithms. These can be divided into three categories:Supervised learning, for example, linear regression, logistic regression, support vector machines, naive Bayes, and decision trees, learn from labeled data.Unsupervised learning, for example, k-means clustering, hierarchical clustering, and anomaly detection. It learns from unlabeled data and identifies patterns from the data.Reinforcement learning, for example, q-learning and deep q-learning, trains an agent to complete a task within a certain environment, receiving rewards and observations from that environment.

It is possible to consider an additional learning type, semi-supervised learning, which receives data that are both labeled and unlabeled.

Finally, deep learning (DL) is a subset of ML in which artificial neural networks are included. Some examples of artificial neural networks are [59]:Feed-forward neural networks;Radial basis functions;Counter propagation;Learning vector optimization.

Besides artificial neural networks, there are other types of neural networks, namely recurrent neural networks (RNN) and convolutional neural networks (CNN) [60].

In ANN (Figure 6a), usually called feed-forward neural networks, the information flows in one direction only, i.e., the flow of information is from the input to the output direction.

RNN (Figure 6b) captures the sequential information present in the input data while making predictions. This leads to a smaller number of parameters to train and, thus, a lower computational cost.

Finally, CNN (Figure 6c) learns the filters automatically, which helps to extract the right and relevant features from the input data. It is usually used for image data, but can be used with sequential data as well.

### Generative AI

Most recently, there has been generative AI. For the applicability of this review, a brief introduction to generative adversarial networks is done. Generative adversarial networks (GANs) consist of a generator network *G* which generates fresh data samples, while a discriminator network *D* determines whether the samples are authentic or not. The discriminator network strives to reliably recognize the fake samples, whereas the generator network continuously seeks to produce better and more realistic samples. The two networks are trained together in a competitive process. In summation, the network plays the zero-sum game of maximizing log(D(x)) and minimizing log(1−D(G(z))), which is equivalent to [61]:(3)$$min_{G}max_{D}V\left(D,G\right)=E_{x\;p(x)}\left[log\left(D\left(x\right)\right)\right]+E_{z\;p(z)}\left[log\left(1-D\left(G\left(z\right)\right)\right)\right]$$

## 3. Feed-Forward Neural Networks

As mentioned previously, neural networks are a numerical tool used in deep learning. Figure 7 shows a scheme of a neural network. Any hidden or output node consists of a non-linear transformation *f* of the values of the nodes in the previous layer. The neuron is also often named as perceptron (Figure 8). This transformation is the weighted sum of the inputs passed through a nonlinear activation function, as shown in Equation (Equation 4):(4)$$z=f\left(b+x\cdot w\right)=f\left(b+\sum_{i=1}^{m}x_{i}w_{i}\right)$$
where *b* is the bias, $x_{i}$ is the value from the node *i* in the previous layer, and $w_{i}$ is the weight corresponding to that node. It is the training process of the network which obtains the values for the weights and bias.

Some examples of activation functions include the sigmoid function,
(5)$$f\left(\Sigma \right)=\frac{1}{1+e^{-\Sigma }}$$
and the hyperbolic tangent function,
(6)$$f\left(\Sigma \right)=\frac{e^{+\Sigma }-e^{-\Sigma }}{e^{+\Sigma }+e^{-\Sigma }}$$
where Σ is, according to Equation (Equation 4), the weighted sum of the nodes from the previous layer [62].

Therefore, The final network is able to do some non-linear transformation and create decision boundaries capable of accurately classifying complex sets of data. Feed-forward neural networks are universal approximators. It is mathematically proven that a network with at least one hidden layer and sigmoid activation functions can approximate any decision boundary [63].

In order to determine the accuracy of the neural network predictions, the mean square error (MSE) calculates the difference between the target vector and the output vector according to Equation (Equation 7)
(7)$$MSE=\frac{1}{m}\sum_{i=1}^{m}{\left(t_{i}-y_{i}\right)}^{2}$$
where $t_{i}$ is the element of the target vector and $y_{i}$ is the prediction made by the neural network for the same inputs’ originating target $t_{i}$. The differences are squared in order to make sure that negative differences are not subtracted from positive ones [62]. Different accuracy metrics can be used, such as the mean absolute error (MAE) and mean absolute percentage error (MAPE).

In neural network training, there are two different propagation moments, forward propagation and backward propagation. In the forward propagation stage, the data go from the input layer to the output layer, thus giving a prediction or classification. The backward propagation step allows training the network, and the data do the reverse transformation in order to determine the error and adjust the weights and bias by minimizing the cost function, which can be the mean square error (MSE) or any other function [62]. The cost/loss function depends on the weights of the neurons of the network. The gradient descent means that the weights are changed in the direction that presents the steepest descent toward a minimum of the cost function, thus achieving a set of network weights leading to a global minimum of the cost function. The best technique to achieve this in feed-forward neural networks (or multi-layer perceptrons) is called backpropagation of error. By determining the neuron’s transformation function’s gradient, the precise weight-adjustment formulas can be determined. Each neuron will have different contributions to the overall error, meaning that some neurons lead to higher errors than others and, as such, the weight adjustment of those neurons should be greater. If the activation function is the sigmoid, the neuron’s contribution to the overall error can be calculated as follows: For a neuron in the output layer:(8)$$\delta_{i}^{\left(1\right)}=y_{i}\left(1-y_{i}\right)\left(t_{i}-y_{i}\right)$$
where $\left(t_{i}-y_{i}\right)$ is the difference between the output *i* and the correct target which is multiplied by $y_{i}\left(1-y_{i}\right)$, a term which is brought to a minimum when $y_{i}$ = 0 or $y_{i}=1$ and brought to a maximum when $y_{i}=0.5$. This way, if this term is minimized, the neuron has a high error contribution and if the term is maximized, the neuron is neutral. For a neuron in the hidden layer:(9)$$\delta_{i}^{\left(2\right)}=h_{j}\left(1-h_{j}\right)\sum_{i}\delta_{i}^{\left(1\right)}w_{ji}$$
thus, the contribution of the weight of a neuron in the hidden layer is calculated by backpropagating the error contribution of the neurons in the output layer through the term $\sum_{i}\delta_{i}^{\left(1\right)}w_{ji}$. Each $\delta_{i}^{\left(1\right)}$ is multiplied by the weight of the link connecting the output neuron *i* to the hidden neuron *j*. Now, from the previously calculated values of $\delta_{i}$, it is possible to calculate the weight update. For a neuron in the output layer, the weight update is:(10)$$w_{ji}^{\left(1\right)}:=w_{ji}^{\left(1\right)}+\mu \delta_{i}^{\left(1\right)}h_{j}$$
and for a neuron in the hidden layer, the weight update is given by:(11)$$w_{kj}^{\left(2\right)}:=w_{kj}^{\left(2\right)}+\mu \delta_{j}^{\left(1\right)}x_{k}$$
where μ is the learning rate and should theoretically be a value between 0 and 1. Each passing of data forward and backward through the network is called an epoch; thus, an epoch consists of using the training data only once. An epoch can consist of one or more iterations depending on whether the batch size is the same as the size of the training data.

There are several algorithms to adjust the weights, which must be chosen according to the type of problem being analyzed and whether the artificial neural network aims at solving a regression problem or a classification problem. The main difference between the two tasks is that a classification problem predicts a label and a regression problem predicts a quantity.

## 4. Accelerating Implant Design with AI/ML

The work of Cilla et al. [64] uses machine learning techniques, namely support vector machines and neural networks, to optimize implant design for minimal stress shielding. The implant design variables are geometrical parameters and the objective variable is the difference in principal strains pre- and post-implant insertion.

Implant design using ANN was also improved in the work of Chanda et al. [65], who focused on primary stability, i.e., before osteointegration starts occurring (which is mostly governed by the micromotion between the bone and the implant). Primary and secondary stability are both vital for the long-term success of total hip arthroplasty, and the lack of biological fixation is a major cause for the aseptic loosening of the implant. The problem had 18 input variables to the network, corresponding to the geometrical parameters of the femoral stem design. The output (aimed at being optimized) was an index of instability. Then, the authors used a genetic algorithm to achieve the optimal design using the modeling from the neural network.

Still, regarding implant design, the work of Ghosh et al. [66] studied bone growth dependency on the microtexture of the implant. The neural network was used to model the level of bone growth according to the geometrical parameters of three different models available in the literature. For the development of an optimized texture for bone growth, a genetic algorithm was used.

In a similar approach to the works of Chanda et al. [65] and Ghosh et al. [66], the work of Roy et al. [67] used ANN to model the dental implant stress and strain as a function of bone and implant geometry parameters, allowing computational time savings by by-passing the FEM analysis stage. Then, a genetic algorithm is used to create the optimal implant geometry.

In summation, artificial neural networks are useful for modeling implant properties as a function of other parameters, such as geometry or bone/tissue properties. Then, using classical optimization approaches, the optimal implant can be developed (Figure 9).

## 5. Accelerating Structural Optimization with AI/ML

Structural optimization is often one of the tools used in the development of prostheses and implants.

Several types of optimization exist with the purpose of finding the best possible configuration for a structure having in mind a certain object and knowing that it must be subjected to some constraints. Structural optimization is in fact an “umbrella term” for a set of techniques aiming at that purpose.

Sizing optimization consists of trying to find, for example, the optimal thickness distribution for a plate or optimizing the thickness of bars in a truss structure, with the goal to minimize physical quantities such as compliance or peak stress. In the sizing optimization problem, the domain of the design model remains the same throughout the optimization process [68].

Shape optimization consists of finding the optimal domain in order to minimize the physical quantities of the objective function [68].

Topology optimization consists of finding the optimal set of features, such as holes, shape and size of holes, and connectivity of the domain, with the same objective of the two previously mentioned techniques [68].

The general topology optimization problem relates to how the material distribution inside the design domain Ω leads to the optimal value of the desired objective function. This material distribution is usually controlled by the density values *x* attributed over the domain. These density values will present values between $\left[0,1\right]$. Thus, the general formulation of the TO problem is as follows:(12)$$\left\{\begin{array}{c} min\;C\left(x\right)=F^{T}u\left(x\right)s.t.\left\{\begin{array}{c} \frac{V(x)}{V_{0}}=f\\ F=K(x)u(x)\\ 0<x_{min}\leq x\leq 1 \end{array}\right. \end{array}\right.$$
which aims at minimizing the compliance function $C\left(x\right)$ and, thus, solving it for maximum stiffness. It is possible to have different objective functions, such as minimal volume for example.

Some of the most common approaches to the solution of the structural optimization problem are the SIMP, ESO (BESO), or level-set methods, which are gradient-based approaches using inverse homogenization theory [68].

Structural optimization is an iterative process where the boundary value problem is solved at each iteration, and is also highly mesh-dependent. Therefore, the use of artificial intelligence can help solve some of those issues.

The early works applying artificial neural networks to structural optimization go back to the 1990s decade, with the following works [69,70,71,72] for example. Most of these early works use neural network architectures consisting of one or more layers of connected neurons and take different approaches to the optimization problem. The development of new network architectures, such as generative adversarial networks, first published in the work of Goodfellow et al. [61], led to the development of some of the most recent works.

In the literature, some more recent examples can be found, such as the work of Chi et al. [73], who used two meshes, a coarse mesh to run fast FEM analysis and a denser mesh to map the design variables. A fully connected deep neural network outputs the sensitivity taking the strains from the coarse mesh and the design variables from the denser mesh. The main highlight of the work is that the training is performed online using the information from previous iterations, by-passing the need for the tedious gathering of data.

Senhora et al. [74] extended the work of Chi et al. [73]. The problem was formulated as a minimal compliance problem through the SIMP model, where two meshes are used: a coarse mesh to calculate the displacement field and a denser mesh with the stiffness distribution. The machine learning, which combines convolutional layers and fully connected layers, takes as inputs the displacement field, obtained with the coarse mesh and the stiffness field from the denser mesh, and presents as output the sensitivity of the objective function, which, in turn, is used to obtain the design variables. The multi-resolution approach is also used by Keshavarzzadeh et al. [75], who solves the expensive TO inexpensively with the coarse mesh and then, through the neural networks, the fine mesh result is obtained.

The approach by Ulu et al. [76] learned optimal structure configuration using neural networks trained with a number of different load cases. Additionally, principal component analysis was used to reduce the dimensionality of data. Sosnovik and Oseledets [77] used convolutional neural networks and employed image segmentation techniques to predict the final optimal configuration from SIMP starting from an intermediate stage of the optimization. Banga et al. [78] then extended the previous work for 3D topology optimization and Lin et al. [79] used CNN to finalize the iterative process starting from an intermediate solution. Another work using the DL approaches for partial solutions is the work of Oh et al. [80], who used generative adversarial networks (GANs) to develop an initial guess, which is then perfected through conventional TO algorithms.

In the work of Yu et al. [81], supervised learning is used to obtain an initial low refinement solution and then, using GANs, the solution is upscaled to a higher refinement. Cang et al. [82] used feed-forward networks and achieved near optimal solutions of topology optimization problems without any iteration, similar to the work of Yu et al. [81]. With a different goal of optimizing heat transfer, the work of Li et al. [83] also consists of two-step optimization, first obtaining a low-resolution solution with GANs, which is refined with a separate GAN. The GAN network was also used by Rawat and Shen [84] to achieve optimized 3D structures. Abueidda et al. [85] used CNN to run topology optimization structures using non-linear material models and non-linear constraints.

White et al. [86] developed a multiscale approach, where a feed-forward neural network with one layer was developed as a surrogate model for the material, whereas the optimization procedure itself was run through the SIMP approach.

One of the most costly steps in structural optimization frameworks is the FEM analysis. The work of Lee et al. [87] replaces the FEM (which calculates the objective function) with a CNN, reducing the computational cost. Rade et al. [88], using both CNN and U-net architectures, also by-pass the FEM step of the optimization procedure. The developed framework has two networks, one for compliance prediction (thus by-passing the FEM) and another one for density prediction (this by-passes the sensitivity analysis step). Qiu et al. [89], using U-net, CNN, RNN, and long short-term memory (LSTM) approaches, developed a framework that by-passes the FEM step by having as input the density result from the previous iteration. Therefore, even though the iterative process still exists, it is much less costly.

In order to reduce computational cost, it is also possible to establish direct mapping between the initial conditions and the final structure configuration. Some works opt for this approach. For instance, Zheng et al. [90] mapped the final structure from the applied loads and boundary conditions, and Yan et al. [91] established direct mapping between the principal stress field at the initial iteration and the final density. Nie et al. [92], using in their work GANs as a core tool, extracted information from the variable fields of the original domain instead of mapping the boundary conditions to run structural optimization. Zhang et al. [93] also had as input in their network the variable fields of the initial structure configuration, and the chosen architecture was a deep CNN. Wang et al. [94] also aimed at real-time topology optimization with U-nets and, thus, trained the network with the results of several topology optimization analyses through an MMC approach. Thus, the iteration process is no longer necessary.

Chandrasekhar and Suresh [95] used the feed-forward neural network properties to model the problem by implementing the SIMP interpolation scheme as the activation functions of the neural network, and, thus, the optimization procedure consists of the minimization of the loss function of the network, similar to conventional procedures where the compliance of the structure is minimized.

Other approaches include the work of Lei et al. [96] with a moving morphable component (MMC) approach to generate training data and using support vector machines (SVR), and K-nearest neighbour (KNN) ML approaches to establish a direct mapping between the design parameters and the optimized topology. In the work of Oh et al. [97], GANs were used to run generative design.

## 6. Combining ML/DL with Homogenization Techniques

Multiscale modeling refers to an approach in which analysis of the material is conducted at one length scale, but the outcomes of the analysis are referent to several properties of the material at another length scale [98].

The use of numerical homogenization techniques allows for significant savings in computational time. Often in composites, it is not necessary and inefficient to model the entire composite’s structure. Instead, only a representative region is chosen to model all the constituents of the composite [99]. This approach can be extended to lattice materials by simplifying the assumption, where the composite presents two or more phases (fiber and matrix) and the lattice will only present one phase, with the rest being a void phase.

The homogenization scheme of a cellular material or lattice is schematically shown in Figure 10. In summary, the porous material is transformed into an equivalent solid material with homogenized properties [100].

FE homogenization can be used in order to predict the effective elastic (as well as the elastic–plastic) properties of the material. This method excels at obtaining the properties of materials with complex microstructures, such as composites [99], even though, in this case, the focus is on lattice materials.

Considering the representative volume element (RVE) Ω, any micro-field f, such as stress or strain within the RVE, can have the following average functions defined:(13)$$\langle f\rangle =\frac{1}{V}\int_{\Omega }f\left(x\right)dx$$
where *V* is the volume of the lattice.

The effective mechanical properties do not depend on the body forces or boundary conditions. Thus, to predict the properties of the material, the following weak-form quasi-static equilibrium differential equation is considered:(14)∇σ=0inΩ

The boundary conditions do not affect the material properties, but they must satisfy Hill’s energy law, which states that the energy on the micro-level has to be the same as the effective energy for the homogenized material:(15)〈σ:ε〉=〈σ〉:〈ε〉

For any point of the RVE, the constitutive model is given as:(16)σ(x)=σ(x,ε(x))

Based on this last relationship, as well as the quasi-static equilibrium differential equation and the boundary condition satisfying Hill’s energy principle, the stress σ(x) and strain ε(x) fields can be obtained through FE analysis, and then, the average values for stress and strain can be computed through:(17)$$\langle \sigma_{ij}\rangle =\frac{1}{V}\sum_{e=1}^{n_{e}}V_{e}\left[\sum_{I=1}^{n_{inte}}\sigma_{ij}\left(y_{I}\right)J\left(y_{I}\right)W\left(y_{I}\right)\right]$$
(18)$$\langle \varepsilon_{ij}\rangle =\frac{1}{V}\sum_{e=1}^{n_{e}}V_{e}\left[\sum_{I=1}^{n_{inte}}\varepsilon_{ij}\left(y_{I}\right)J\left(y_{I}\right)W\left(y_{I}\right)\right]$$
where $n_{e}$ is the number of elements in the model, $n_{inte}$ is the number of integration points in the element *e*, and $W(y_{I})$ is the weight of the integration point. $\sigma_{ij}\left(y_{I}\right)$ and $\varepsilon_{ij}\left(y_{I}\right)$ are the stress and strain, respectively, evaluated at the integration point $y_{I}$.

The average values, which consist of a volume average of the properties along the material, can then be used to calculate the effective elastic stiffness tensor C and the effective stress tensor for elastic–plastic analysis.

There is a growing number of examples in the literature combining homogenization approaches with neural networks.

Some works combine ML/DL approaches with homogenization techniques, such as the work of Alwattar and Mian [100], who used feed-forward neural networks to calculate the equivalent properties of body-centred-cubic unit cells, which were then validated using an equivalent solid model and compared to experimental testing. Similarly, in the work of Koeppe et al. [101], long short-term memories (LSTM) are used to obtain the equivalent properties of lattice unit cells and validated experimentally. Additionally, Kollmann et al. [102] used a CNN model to predict the optimal design for bulk and sheer modulus maximization and Poisson’s ratio minimization, which, in summary, consists of the reverse homogenization problem. Settgast et al. [103] used neural networks to model the anisotropic elastic–plastic behaviour of foam materials.

Two approaches can be identified for the combination of both techniques. The first is a direct approach where the neural network acts as a surrogate model for the mechanical properties of the meta-material. This is relevant since it can be integrated with structural optimization algorithms for material homogenization to accelerate the iterative procedure. The second is a reverse approach where the neural network capabilities of modelling high-dimensionality functions are used to predict some optimal geometry from the desired mechanical properties. This is especially relevant in tissue engineering where the mechanical properties should be finely tuned to mimic the behaviour of the biological tissue it is replacing. Wang et al. [104] used ANN, namely feed-forward neural networks, to predict the stress–strain curves of TPMS cells and, through an optimization algorithm, TPMS with specific characteristics can be achieved from a desired stress–strain curve. In the work of Ma et al. [105], a neural network is used to accelerate the homogenization step in topology optimization. Kim et al. [106] predicted the anisotropic elastic properties of the unit cell, necessary for topology optimization from a neural network whose inputs were the geometric parameters of the unit cell. White et al. [86] also used a neural network to predict the elastic properties of the material for topology optimization. Zheng et al. [107] used the constraints from the sensitivity analysis run in the macroscale, and then the neural network output the stiffness tensor. In the work of Zhu et al. [108], the neural network correlates the elastic tensor with the radii of struts in the lattice material. Black and Najafi [109] uses a neural network as a surrogate model for the homogenization and sensitivity analysis step of the topology optimization procedure.

All of the previously mentioned approaches follow a direct path; on the other hand, Challapalli et al. [110] presents a neural network aimed at obtaining an optimal geometry that follows the elastic property requirements. In this work, a GAN optimizes the lattice cell based on boundary conditions. In addition, Zheng et al. [111] uses a conditional GAN to create a structure presenting a specific Young’s modulus and Poisson ratio. Kollmann et al. [102] presents a DL model which creates unit cells for the maximization of shear or bulk modulus or the minimization of Poisson’s ratio, which obtains auxetic structures. Garland et al. [112] achieves Pareto optimal unit cells, with a CNN trained as a surrogate model for the stiffness of the unit cell. Then, using the trained CNN, a genetic algorithm optimizes the unit cell geometry. In a similar trajectory, Ji et al. [113] also presents a neural network to predict the homogenized elastic tensor, while a genetic algorithm optimizes the unit cell geometry. Finally, Patel et al. [114] develops a neural network that creates the optimal unit cell to fit the macroscale topology problem.

## 7. Biomaterial Charcterization Combined with AI/ML/DL

This section shows some examples of how the techniques which were mentioned above can be used in material characterization beyond homogenization techniques. This will refer to material parameters, mechanical properties, and processing parameters. Additionally, since flexible sensors are a common application of cellular materials, some examples combining AI with those sensors are shown in this section.

### 7.1. Material Parameter and Property Characterization

Material properties and parameters and process parameters can be studied or obtained through the use of AI/ML/DL techniques without the combination of homogenization techniques. These properties are also important for the study of biomaterials.

Shojaei et al. [115] used gene expression programming to obtain the hardness, Young’s modulus, apparent porosity, and relative density of Ti/carbonated HA composites. Wang et al. [116] predicts the elastic modulus, as well as the cell-material interaction and cell permeation of scaffolds, with several machine learning techniques such as Gaussian process regression, neural networks, kernel ridge regression, and support vector regression.

Regarding the influence of process parameters, Reddy et al. [117] modeled the properties of electrospun PCL scaffolds as a function of the electrospinning parameters through neural networks.

### 7.2. Flexible Sensors

Due to the high sensitivity of cellular materials to detect pressure signals, AI can be combined with cellular materials to form flexible sensors. In these cases, pressure signals can be interpreted through AI algorithms. For this case, AI/ML/DL techniques are not employed in the study of the cellular material, but, instead, are applied in further stages of the application. Some works illustrating applications of cellular materials in flexible sensors are, for example, the work of Sun et al. [118], with a flexible biomaterial device based of corn-starch; Yang et al. [119] used 1T molybdenum disulfide within polydimethylsiloxane foams as a spacer for their sensor; and Yu and co-workers, with a GSB, enhanced a three-dimensional graphene composite strain sensor capable of a stretching ratio of 50%, high sensitivity, and a response time of 128 ms, Yu et al. [120]. Cao et al. [121] developed 3D-printed graphene thin and thick wall cellular microstructures which achieved low detection limits, a wide detection range, and high sensitivities. Fang et al. [122] used cellular PP as the piezoelectret and, in their work, pattern-recognition techniques were employed to detect different hand and wrist movements, thus showing how AI can be employed in data analysis of the sensor signal.

## 8. Discussion and Conclusions

In summary, the studied approaches increase the speed of the topology optimization procedure by starting from partially converged solutions or by working on coarser mesh scales and using DL to determine the refined solution. Moreover, another main achievement of the mentioned approaches is to by-pass the costly iterative process, which is common in structural optimization problems. Additionally, it was verified that besides using as input the loads and boundary conditions of the structure, relevant information can be extracted from the variable fields of the initial configuration. Regarding architectures, GAN using CNN layers are some of the most common networks used in structural optimization, as well as U-net architectures. It is found that some works (sometimes based in image recognition approaches) aim at recognizing features in the structure, whereas other works take a different approach and aim at developing a map between initial conditions and final structure configuration. However, as in Lynch et al. [123], other authors intended to accelerate other costly steps of the optimization procedure, such as the algorithm parameters.

The ML approaches chosen for the previously mentioned works in Section 5 are summarized in Table 8. It is interesting to point out that research works in the field of biomechanics tend to use feed-forward neural networks, whereas works in the field of structural optimization show a wider range of approaches, often combining several architectures.

Within SO, a wider variety of frameworks appeared. The most costly steps in SO frameworks are the sensitivity analysis and the FEM/structural analysis. This can be accelerated by direct mapping from initial conditions, removing the need for the iterative process and the number of analyses. Additionally, the optimization itself is embedded in the network, as the minimization of the cost function takes the physical meaning of minimizing the objective function, which, in this case, is the compliance function.

Finally, implant design and structural optimization can be accelerated by the combination of ML/DL techniques at the homogenization stage. It was verified generally that two surrogate model solutions are used: one in which the homogenization is conducted by the neural network, and another in which the neural network is used to map ideal unit cells from the required elastic properties (to avoid stress shielding, for example).

## Figures and Tables

**Figure 1 materials-16-03946-f001:**
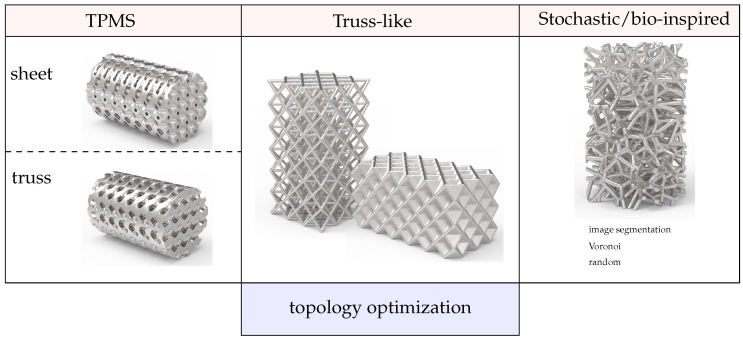
Scaffold topology classification: TPMS sheet-like (for example, [7]); TPMS strut-like (examples in [22]); truss-like or CAD-based (examples in [23]); truss-like from topology optimization to fit some stiffness/strength requirements (examples in [21]); stochastic scaffold (examples in [24]).

**Figure 2 materials-16-03946-f002:**
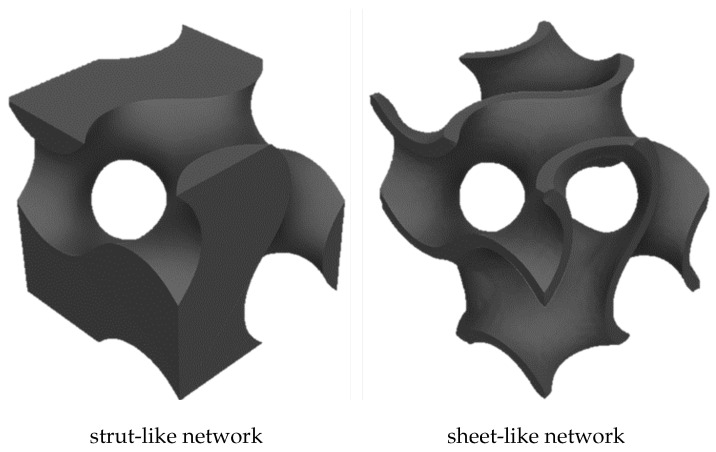
Strut-like TPMS and sheet-like TPMS.

**Figure 3 materials-16-03946-f003:**
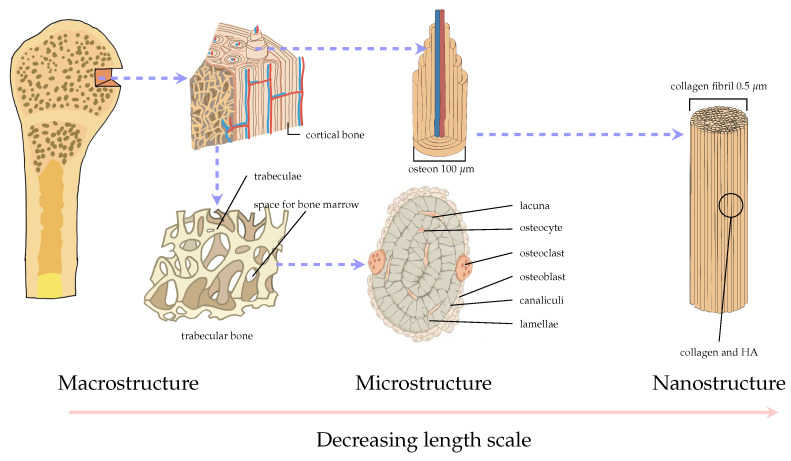
Hierarchical structure of bone, dashed arrows indicate length scale changes.

**Figure 4 materials-16-03946-f004:**
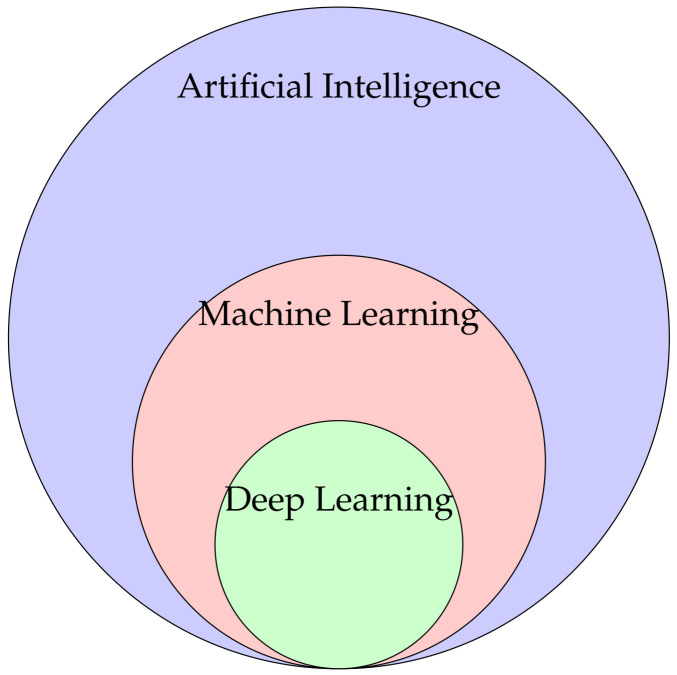
Applicability of the terms: artificial intelligence, machine learning, and deep learning.

**Figure 5 materials-16-03946-f005:**

Steps for the development of a machine learning model.

**Figure 6 materials-16-03946-f006:**
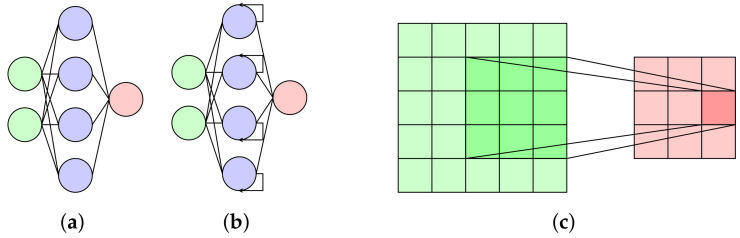
Types of artificial neural networks: (**a**) feed-forward neural network; (**b**) recurrent neural network; (**c**) convolutional neural network.

**Figure 7 materials-16-03946-f007:**
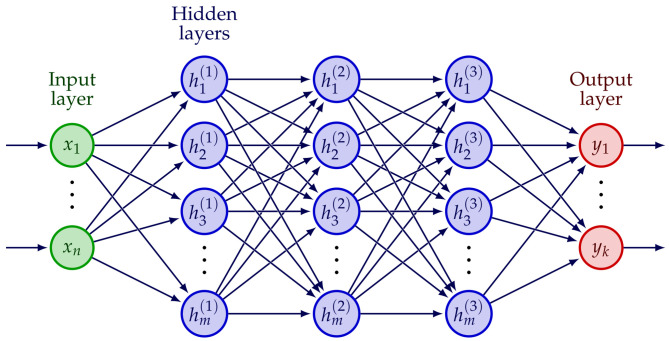
Artificial neural network architecture.

**Figure 8 materials-16-03946-f008:**
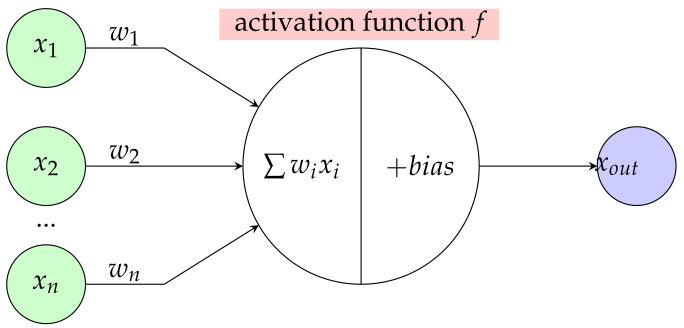
Nonlinear transformation at a neuron or perceptron.

**Figure 9 materials-16-03946-f009:**
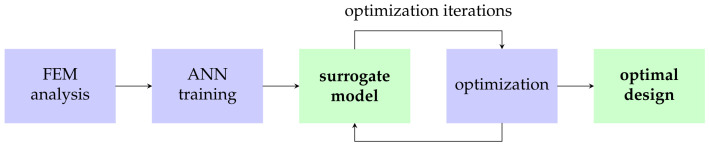
Implant design methodology combining optimization and neural networks.

**Figure 10 materials-16-03946-f010:**
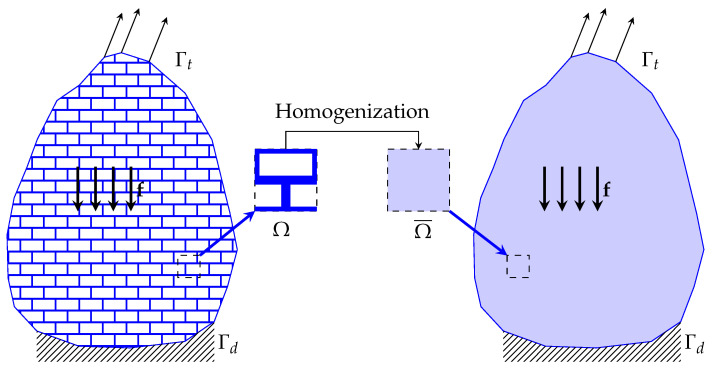
Homogenization concept of a lattice structure.

**Table 1 materials-16-03946-t001:** Mechanical properties collected through the review works of Bobbert et al. [7] and Wang et al. [6].

	Miscellaneous	Vertebra	Proximal Tibia	Greater Trochanter	Femoral Neck	Femur	Proximal Femur
Young’s modulus [MPa]	10–1570 (trab.) 5000–23,000 (cort.)	90–800 (trab.)	200–2800 (trab.)	200–1500 (trab.)	750–4500 (trab.)	389 ± 270 (trab.) 16,700 (cort.)	441 ± 271 (trab)
Strength [MPa]	164–200 (cort.)	0.56–3.71 (trab.)	-	-	55.3 ± 8.6 (trab.)	122.3 (cort.)	-
Poisson’s ratio	0.4 ± 0.16 (cortical, longitudinal) 0.62 ± 0.26 (cortical, transverse)	-	-	-	-	-	-

**Table 2 materials-16-03946-t002:** Comparative summary of additive manufacturing processes, adapted from [25].

ASTM Classification	Description	Tecnology	Materials	Energy Source	Advantages	Disadvantages
Material Extrusion	Material is selectively extruded through a nozzle or orifice	Fused deposition modelling (FDM)/ Fused filament fabrication (FFF)	Thermoplastic; Reinforced thermoplastic (with p.e. metal, wood, ceramic, fiber)	Heat	Equipment can be inexpensive (in relative terms); multi-material printing	Limited detail; Poor surface finishing
		Contour crafting	Construction materials			
Powder bed fusion	Thermal energy selectively fuses/sinters regions of a powder bed	Selective laser sintering (SLS)	Polymers; Polyamide	Laser	High resolution; Abilty to print fully dense parts; Good mechanical properties of printed parts	Powder handling and recycling; Support and anchor structure
		Direct metal laser sintering (DMLS)	Metals			
		Selective laser melting (SLM)				
		Eletron beam melting		Eletron beam		
Vat photopolymetrization	Liquid photopolymer in a vat is selectively cured by light activated photopolymer	Stereolitography (SLA)	Resins; reinforced resins	Laser	Building speed (especially DLP); High resolution	Overcuring, scanned line shape; High cost
		Digital light processing (DLP)		UV light/Laser (projected)		
Material jetting	Droplets of build material are selectively deposited.	Polyjet/inkjet printing	Photopolymer	Heat/Light (laser, UV)	Multi-material printing; surface finish; resolution	Mechanical properties
Binder jetting	A bonding liquid agent is selectively deposited to join powder materials	Binder jetting	Polymers; Ceramics; Metals	Heat	Printing of fully coloured objects; Material range	Post-processing; High porosity of printed parts
Sheet lamination	Sheets of material are bonded	Laminated object manufacturing (LOM)	Paper; Plastic film; Sheet metal; Ceramic tape	Laser	Surface finish; Low cost	Post-processing
Directed energy deposition	Focused thermal energy is used to fuse materials by melting as these are being deposited	Laser engineerined net shaping (LENS)	Metals	Laser	Can be used to repair damaged parts such as dies and molds; Functionally graded parts	Post-processing
		Electronic beam welding (EBW)				

**Table 3 materials-16-03946-t003:** Summary of the mechanical properties (compressive) of some metallic scaffolds found in the literature.

	Process	Geometry	Material	Porosity	Pore Size (μm)	E (MPa)	Strength (MPa)
[27]	SLM	lattice (CAD-based)	Ti-6Al-4V	23.6–91.7%	-	250–19,000	25–1225
[31]	SLM	lattice (CAD-based)	Ti-6Al-4V	65%	1804–2187	8.61–10.78	75.1–87.26
[28]	SLM (LPBF)	BCC	Zn-xMg alloy	80.50%	∼ 800	170–650	7.34–19.12
[29]	SLM (LPBF)	lattice (CAD-based)	Zn-xWE43	45% (67% designed)	600	950–2540	22.9–73.2
[30]	SLM	primitive strut	Fe-35Mn, Fe-35Mn-1Ag	60%	500	14.76, 14.59	41, 75
[32]	SLM (LPBF)	sheet (gyroid, primitive, diamond, IWP)	Ti-6Al-4V	54–60%	710–900 (designed)	4100–6700	180–250
[26]	SLM	gyroid sheet	316L stainless steel	75.1–88.8%	500–1300	308–1116 (1026–2004 FEM)	6.9–29 (15.9–43 FEM)
[33]	SLM	sheet (gyroid, diamond, FKS)	Ti-6Al-4V	49.5–75.40% (45.37–72.99% actual)	400–800	2000–5100	86.7–264.2
[34]	EBM	gyroid strut	Ti-6Al-4V	82–85%	850–1270	637–1084	13.1–19.2
[35]	SLM	primitive strut	Fe-35Mn	42%	400	33.5	89.2

**Table 4 materials-16-03946-t004:** Summary of the mechanical properties (compressive) of some polymeric scaffolds found in the literature. (* means approximated values).

	Process	Geometry	Material	Porosity	Pore Size (μm)	E (MPa)	Strength (MPa)
[36]	FDM	random	PLA	23.25–60.34%	400–6006	25.3–27.8, 1300–1600	-
[37]	FDM	FCC, BCC, cubic, tetragonal	PLA+hydrogel shell	47.2–89.5%	-	100–500	2–18
[38]	SLS	gyroid strut, image-based	nylon	gradients: (58–70)%, (30–70)%, (50, 80)%	-	51–171	4–7 *
[38]	SLA	gyroid strut, image-based	resin	gradients: (58–70)%, (30–70)%, (50, 80)%	-	-	-
[23]	FDM	CAD-based	PLA	-	350	183.62 (213.21 numerically)	-
[39]	PolyJet	CAD-based	Photopolymer	60%	2500	113 (129 numerically)	4.3 (4.7 numerically)
[40]	FDM	gyroid sheet, primitive sheet	PLA	61.4–61.7%	-	264,221	8.68, 7.06
[41]	FDM	Voronoi	PLA	71%	4000–11,800	-	-
[22]	FDM	(gyroid, diamond, Schwarz) strut	PLA	35–65%	600–1400	82–690	4–34

**Table 5 materials-16-03946-t005:** Summary of the mechanical properties (compressive) of some ceramic scaffolds found in the literature.

	Process	Geometry	Material	Porosity	Pore Size (μm)	E (MPa)	Strength (MPa)
[4]	LCM	primitive, sheet	Alumina	>50%	500–1000	2000–5500	11–133.5
[24]	DLP	voronoi	Zirconia(80%)+HA(20%)	35.53–61.75%	-	103,500–174,180	9.95–19.40
[42]	DLP	CAD-based	Graphene oxide (0–0.4%)+HA	43.7–70%	346.7–421.7	-	0.25–1.52
[43]	Extrusion	CAD-based	HA+β tricalcium phospahte	41.32–56.12	350–733	19,018–30,948	22.5–49
[44]	Extrusion, DLP	gyroid, diamond, CAD-based	Borosilicate glass	52–78%	-	-	0.9–12.2
[46]	projection-based	CAD-based	HA	54.6	500	380	6.18
[47]	DLP	image-based	silica doped-HAp	-	-	-	3.93–12.94
[48]	litography-based	CAD-based, octa, cube	HA	-	-	254.87–287.57	2.28–5.6
[45]	DLP	BCC	HA	49.8% (54.6% designed)	300–600	970	15.25
[49]	litography-based	CAD-based	β-tricalcium phosphate	5.58–59.36%	200–800	-	1.4–67.6
[50]	DLP	CAD-based (octagon+rhombic)	Alumina	62.80–80%	566–1000	500–4000	6.50–30

**Table 6 materials-16-03946-t006:** Summary of the mechanical properties (compressive) of some composite scaffolds found in the literature.

	Process	Geometry	Material	Porosity	Pore Size (μm)	E (MPa)	Strength (MPa)
[51]	FDM	CAD-based	HA(0,20,40%)+PEEK	69.3–70.6%	800–1600	-	-
[53]	FDM	CAD-based	HA(0,20,40%)+PEEK	64–70%	400–500	240–410	16–27
[54]	FDM	CAD-based	HA+PLA	-	200–450	-	-
[55]	FDM	CAD-based	HA(0,20,40%)+PEEK	47.3–87.8%	200–2000	50.6–624.7	2.2–35.2
[56]	FDM	CAD-based	PLA+316L, PLA+Fe	47% (designed)	780–880	178–1390	15–36
[57]	FDM	CAD-based	PLA/PCL (+Bre, +Bre+Sr)	48–58%	400	33.98–105.79	5.37–14.29
[58]	FDM	CAD-based	PCL+Mg-1Ca(0,5,10,20%)	50%	400	55.9–96.8 (0% and 10%)	5.7–10.3

**Table 7 materials-16-03946-t007:** Difference between artificial intelligence, machine learning, and deep learning.

	Artificial Intelligence	Machine Learning	Deep Learning
Description	Automation of tasks, learning from patterns	Implementation of AI, algorithms that can learn from data (supervised learning, unsupervised learning, reinforcement learning)	Subset of machine learning; it is a way to implement machine learning which can be used for complex tasks such as classification, feature extraction, etc.
Examples	Computer programs	Decision trees, naïve Bayes, random forest, support vector machine, K-nearest neighbor, K-means clustering, Gaussian mixture model, etc.	Neural networks (feed forward, convolutional, recurrent, etc.)

**Table 8 materials-16-03946-t008:** Summary of ML/DL tools used in the literature survey on biomechanics (implant design) (B) and structural optimization (SO) using ML/DL approaches.

		ML/DL Tool/Approach													
Reference	Outline	CNN	FFNN 1 Layer	FFNN +1 Layer	DDNSM	GAN	U-Net	Res-Net	SE-Res Net	SVR	KNN	Unsupervised K-Means	Bayesian Optimization	LSTM	RNN
[64]	B		x												
[65]	B		x												
[66]	B		x												
[67]	B		x												
[74]	SO	x		x											
[73]	SO			x											
[76]	SO		x												
[77]	SO	x													
[78]	SO	x													
[75]	SO				x										
[124]	SO	x				x									
[92]	SO					x	x		x						
[83]	SO					x									
[93]	SO	x					x								
[125]	SO					x									
[85]	SO						x	x							
[96]	SO									x	x				
[97]	SO					x									
[82]	SO			x											
[126]	SO														
[79]	SO	x					x								
[84]	SO	x				x									
[127]	SO	x				x									
[128]	SO											x			
[80]	SO					x									
[81]	SO	x				x									
[129]	SO	x													
[86]	SO		x												
[87]	SO	x		x											
[95]	SO			x											
[123]	SO												x		
[88]	SO	x					x								
[90]	SO						x			x	x				
[91]	SO	x													
[94]	SO						x								
[89]	SO	x					x							x	x

## Data Availability

Not applicable.

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
