# Peer review of "Advances in Computational Techniques for Bio-Inspired Cellular Materials in the Field of Biomechanics: Current Trends and Prospects"

_materials, 2023, doi:10.3390/ma16113946_

Round 1
Reviewer 1 Report
Comments and Suggestions for Authors
The review paper titled “Advances in computational techniques for bio-inspired cellular materials in the field of biomechanics: current trends and prospects” by A.I. Pais, and associates worked out on orthopedic biomechanics research such as implant and scaffold design. Review work is good, well drafted, presented novel information in computational model in biomechanics. At some points english/grammar, sentence consistency and needed accuracy. From our side following points are addressed that could be taken positively:
Comment 1. Enrich your introduction part with some more on current orthopedic biomechanics.
Comment 2. Authors mentioned 1.1. Meta-materials. Under this heading authors should explain what is meta-material for the beginners and quote few examples of met-materials.
Comment 3. While explaining additive manufacturing give some background on to them for better understanding.
Comment 4. Add few more sentences on 1.3.2. Polymers and 1.3.4. Composites.
Comment 5. Authors explain the differences in Artificial intelligence, machine learning and deep learning and give more information under heading 2. Artificial intelligence, machine learning and deep learning for reader’s easiness.
The manuscript could be considered for publication only after revision. Moreover, I am glad to review a revision of this manuscript if necessary.
Minor editing of English language required
Author Response
Dear reviewer 1,
Please find the answers to all the questions in the attached file.
Best regards,
The authors

Reviewer 2 Report
Some comments:
1. Discusses advances and offers an overview in computational techniques for bio-inspired cellular materials in the field of biomechanics. The article provides a platform for future research and development in this area using AI.
2. May provide an additional systematic classification of cellular materials into:
i)Hierarchical
ii) Gradient
iii) Origami-based
iv) Morphing
v) Biomimetic
3. Mathematical background of perceptron, activation and loss functions are well expressed. For completeness, a brief paragraph/equations on gradient descent or related optimisation functions for backpropagation is required.
4. Scanty details on generative AI. A couple of paragraphs introducing generative deep learning models will considerably broaden the readability.
5. In addition to the sections on "Accelerating implant design" and "Accelerating structural optimisation", the following integral parts are omitted from the current manuscript:
i) Material characterisation: Use of AI in determining properties of different biocellular materials.
ii) Feedback control systems: Monitoring mechanical properties of materials by sensors and adjustments.
6) The section on "combining ml/dl with homogenisation techniques" could be expanded on data-driven homogenisation, parameter identification of materials, uncertainty estimation in the homogenisation of materials.
Mostly okay
Author Response
Dear reviewer 2,
Please find the answers to all the questions in the attached file.
Best regards,
The authors

Reviewer 3 Report
After reviewing this manuscript it is the opinion of this reviewer that the article represents a comprehensive and well organized review of the current state of material design as it applies to orthopedic medicine. The supplied graphics convey the overarching premise of the article well, though with regards to figures that contain graphics adopted from other published works a comment to this effect and acknowledgement that the correct permissions were obtained where needed would be helpful. Apart from this it is the opinion of this reviewer that the work offers a valuable insight into this topic that will serve as an important resource for readers operating in a similar field, and as such the paper should be accepted for publication in its current form permitting that the above acknowledgements be put in place where appropriate.
Author Response
Dear reviewer,
The authors acknowledge your comments. Thank you very much.
Best regards,
The authors
Round 2
Reviewer 2 Report
can be acccpted
minor